# 🌰ACORN: Aspect-wise Commonsense Reasoning Explanation Evaluation

**Ana Brassard,**[αβ] **Benjamin Heinzerling,**[αβ] **Keito Kudo,**[βα]
**Keisuke Sakaguchi**[βα] **& Kentaro Inui**[γβα]
[α]RIKEN, [β]Tohoku University, [γ]MBZUAI

{ana.brassard, benjamin.heinzerling}@riken.jp
{keisuke.sakaguchi, kentaro.inui}@tohoku.ac.jp
keito.kudo.q4@dc.tohoku.ac.jp

## Abstract

Evaluating the quality of free-text explanations is a multifaceted, subjective, and labor-intensive task. Large language models (LLMs) present an appealing alternative due to their potential for consistency, scalability, and cost-efficiency. In this work, we present 🌰ACORN, a new dataset of 3,500 free-text explanations and aspect-wise quality ratings, and use it to evaluate how LLMs rate explanations. We observed that larger models outputted labels that maintained or increased the inter-annotator agreement, suggesting that they are within the expected variance between human raters. However, their correlation with majority-voted human ratings varied across different quality aspects, indicating that they are not a complete replacement. In turn, using LLMs as a supplement to a smaller group of human raters in some cases improved the correlation with the original majority labels. However, the effect was limited to cases where human raters were scarce, and an additional human rater had a more pronounced effect in all cases. Overall, we recommend against using LLMs as a complete replacement for human raters but encourage using them in configurations that end with targeted human involvement.

🔗 a-brassard/ACORN

## 1 Introduction

Natural language processing systems that not only generate correct output, but also provide an explanation (Miller, 2019) of why that particular output is correct, are desirable for several reasons, such as increasing trustworthiness (Floridi, 2019), compliance with "right to explanation" laws (e.g., European Parliament & Council of the European Union), increasing interpretability (Jacovi & Goldberg, 2020, but cf. Lipton, 2018 on the caveats of post-hoc explanations), as well as system improvement and knowledge discovery (Adadi & Berrada, 2018). However, this immediately raises the question of how to evaluate the plausibility of system-generated explanations in an efficient and effective manner.

Since explanations are typically free-form text, automatic evaluation of explanations suffers the well-known, but as of yet unresolved, weaknesses of automatic evaluation measures (Celikyilmaz et al., 2021), while human evaluation is characterized by low scalability, high costs, subjectivity, and inconsistency (Hartmann & Sonntag, 2022). LLM-based evaluation presents an appealing alternative due to its potential high scalability, low cost, and consistency.

Here, we investigate whether LLMs (Brown et al., 2020; OpenAI, 2023, i.a.) can serve as a viable alternative approach to automatically evaluate system-generated explanations. To verify the feasibility of this approach, we created 🌰ACORN, a new dataset of 3,500 textual explanations with aspect-wise human ratings of their quality, a first of its kind, and used it to evaluate whether LLMs are aligned with human judgments (Figure 1).

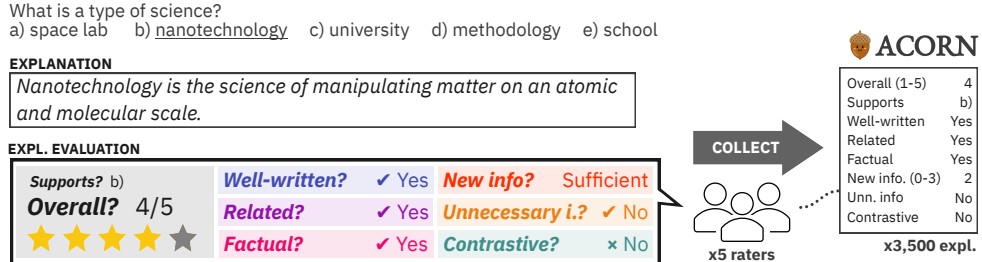

Figure 1: We collected aspect-wise human ratings for 3,500 textual explanations for commonsense reasoning benchmarks. We compared these against ratings from large language models (LLMs) to evaluate their alignment with human judgments.

Specifically, we first considered whether LLMs' labels deviate from what would be expected from human raters. We compared inter-annotator agreement between all-human annotation (five raters) and when one of the raters is replaced by an LLM. We found that stronger models maintained or marginally increased inter-annotator agreement in most cases, suggesting that it may have potential as a replacement for human judgments.

To verify this, we considered two scenarios: one where the LLM replaces the full human evaluation, and one where it is used only as an additional rater. In other words, we compared LLMs to humans *collectively* and *individually*.

With the best-performing model, Spearman's rank correlation between majority-voted gold labels and LLM-generated ones ranged between 0.54 and 0.93, depending on the aspect, averaging 0.72. This indicates that the LLM's labels are not entirely in line with human judgments, but they are not entirely random either, and their usefulness may depend on the specific evaluation criteria and usage case.

As for using the LLM as a single rater, we considered whether, in a limited-budget scenario, it would be beneficial to use an LLM as an *additional* rater. Specifically, we compared whether the majority-voted labels of a reduced set of raters and with an added LLM had a higher correlation with the original majority-voted labels. In cases with three or more raters, the addition of LLMs either had no effect or was detrimental. In cases with one or two raters, the addition of LLMs marginally improved the correlation with the original majority-voted labels, however, still less so than the addition of a human rater.

In summary, we quantified the consequences of using LLMs as a replacement or addition to human evaluation. We conclude that they are an imperfect approximation of human majority votes but still within the expected variance that comes with a subjective task, with some potential benefit when human raters are scarce. Thus, we recommend against using LLMs as a complete replacement for human raters, but encourage using them in configurations that end with targeted human involvement, such as extensively using LLMs in development or filtering stages and human experts for final testing or evaluation.

## 2 Building 🌰Acorn: Evaluation Criteria and Data Sources

In a typical commonsense reasoning test, a model selects the most plausible answer for a multiple-choice question. In a predict-and-explain setting, the model additionally generates an explanation to justify the selected answer, where we encounter the challenge of evaluating these explanations. Thus, we first define a set of rating criteria (§2.1) and collect human ratings for a selection of existing and newly collected textual explanations (§2.2, §2.3). See Appendix A for more details on the dataset, including label distributions, data source-wise average ratings, and examples.

| Criterion | Description | Label Choices |
|---|---|---|
| Supports | Which answer does it justify? | a), b), ..., none |
| Overall | How good is the given explanation, overall? | 1 to 5 stars |
| Well-Written | Coherent, grammatically correct, fluent? | Yes, No |
| Related | Relevant to the Q and A? | Yes, No |
| Factual | Stated facts are generally true? | Yes, No, N/A |
| New Information | How much *new* information to support the ans.? | None, Some, Sufficient, Ample |
| Unnecessary Info. | Any unnecessary statements? | Yes, No |
| Contrastive | Clearly shows the difference between the ans.? | Yes, No |

Table 1: Explanation rating criteria used in this study. The first two criteria target consistency and general explanation quality, while the rest covers specific quality aspects.

## 2.1 Rating Criteria

We defined a set of criteria to target *surface-level*, *information/content-level*, and *structural* aspects of explanations. We also included criteria to capture *(un)faithfulness* and an *overall* rating, the latter intended to implicitly capture any other aspects considered by the raters. We defined these criteria based on common practices in natural language generation evaluation (Howcroft et al., 2020), known challenges of free-text explanations (Lipton, 2018; Rawte et al., 2023), and insights from social sciences (Miller, 2019). Table 1 summarizes the criteria. The criteria largely aligns with the fine-grained analysis conducted by Wiegreffe et al. (2022).

**Supports** assesses *which* answer the explanation supports, intended to be cross-referenced with the predicted label. A mismatch between the predicted label and the supported answer indicates a lack of faithfulness, i.e., the explanation does not reflect the model's reasoning for the label.

**Overall** is a holistic assessment of the explanation, capturing any potentially informative or useful aspects that we have not explicitly covered. We encouraged workers to consider this criterion *independently* of the other criteria, and provided general guidelines for each star rating to ensure a consistent understanding.

**Well-Written** is a catch-all criterion to assess the surface-level quality of the explanation, combining criteria such as fluency, coherence, and grammaticality.

**Related** assesses the relevance of the explanation to the question and answers.

**Factual** evaluates the truthfulness of the statements in the explanation, if any, regardless of their relevance.

**New Information** assesses the extent to which the explanation provides new information beyond the question and answers. Workers were given the choice of none for a complete lack of new information, some for a partial addition, sufficient for a satisfactory amount of new information, and ample for highly informative explanations.

**Unnecessary Information** assesses the extent to which the explanation includes irrelevant information. We included this criterion to capture the challenge of generating *minimal* explanations.

**Contrastive** assesses whether the explanation contrasts the correct answer with the predicted answer.

## 2.2 Source Datasets

🌰ACORN contains ratings for a diverse set of existing, newly-collected, and generated explanations. Our choice covers two commonsense reasoning benchmarks and their respective explanation datasets (Figure 2). From each, we selected a random subset of 500 explanations for rating, as well as an additional 500 samples of fluency-improved versions, resulting in a total of 3,500 explanations. The fluency-improved subset is included to prevent fluency from becoming a superficial signal, since well-written explanations typically also have high scores in all other aspects. With five raters and eight criteria per sample, this amounts to 140k ratings in total.

Figure 2: We collected general and aspect-wise ratings for human-written, LLM-improved, better human-written, and LLM-generated explanations, for BCOPA and CommonsenseQA, respectively.

Specifically, as the target commonsense reasoning benchmarks, we selected BCOPA (Kavumba et al., 2019)[1] and CommonsenseQA (Talmor et al., 2019) based on the simplicity of their tasks and availability of large-scale explanation datasets (Wiegreffe & Marasović, 2021). Below are the respective datasets we used to source candidate explanations.

**CoS-E** (Rajani et al., 2019) A widely-used explanation dataset for CommonsenseQA, albeit notoriously uninformative to humans (Nauta et al., 2023). A subset is processed through GPT-3.5[2] for fluency improvement (500 samples + 250 fluency-improved versions)

**ECQA** (Aggarwal et al., 2021) An improved version of explanations for CommonsenseQA, aligning with our criteria for well-formed explanations. (500 samples)

**Generated explanations for CommonsenseQA.** Additional high-quality explanations generated by prompting GPT-3.5 to solve a subset of CommonsenseQA, though potential issues like irrelevant information were noted. (500 samples)

**COPA-SSE** (Brassard et al., 2022) Explanations for BCOPA with a subset processed through GPT-3.5 for fluency improvement. Since COPA-SSE already contains overall quality ratings, we selected a random sample of 250 questions and used each question's top-rated and bottom-rated explanation. (500 samples + 250 fluency-improved versions)

**Crowdsourced explanations for BCOPA.** ECQA's counterpart for BCOPA; a new set of hand-written explanations, carefully crafted for contrastiveness and thoroughness. (500 samples)

**Generated explanations for BCOPA.** Similarly to CoS-E, we prompted GPT-3.5 to solve BCOPA questions. (500 samples)

### 2.3 Rating Collection

We crowdsourced ratings for the explanations in 🌰ACORN using Amazon Mechanical Turk (AMT). Each rater was required to pass a qualification test, after which they were asked to participate in trial rounds, during which we addressed several clarity issues in the guidelines. The final pool was hand-picked based on their responses, resulting in 28 participants. See Appendix B for more details on the rating collection process. In experiments which use aggregated labels, they were produced with a majority-vote mechanism, with ties broken using the better label.

## 3 Can LLMs Replace Human Raters?

We answer this question in three steps: we first measure LLMs' divergence from expected human label variance (§3.2), then observe the results when using it to *completely* replace hu-

---

[1]Balanced COPA; a superset of COPA (Gordon et al., 2012) with added "mirrored" questions that flip the correct label, i.e., the originally incorrect choice becomes correct.

[2]`text-davinci-003`; The model was instructed to only improve the fluency and was not given any additional context that may encourage improving the content, e.g., by supplementing related information.

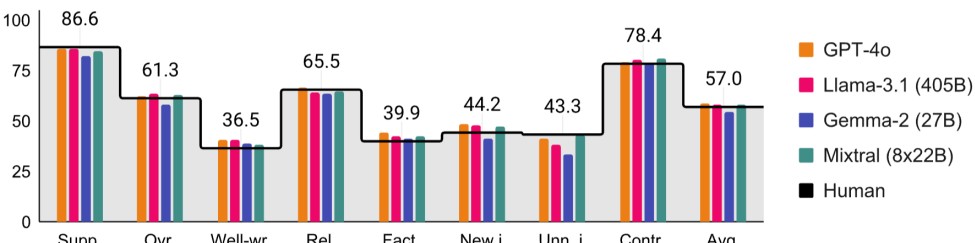

Figure 3: Inter-annotator agreement (Krippendorff's $\alpha$, $*100$ for legibility) between human raters (shaded area) and with the LLM's rating replacing a random rater.

man evaluation (§3.3), and when using it as an *additional* rater instead (§3.4). All experiments follow the settings described in Section 3.1.

## 3.1 Experimental Settings

**Models.** We compared four contemporary API-enabled LLMs, namely GPT-4o (OpenAI, 2024), Llama-3.1 (405B) (Team, 2024), Gemma-2 (27B) (Gemma Team, 2024), and Mixtral (8x22B) (Jiang et al., 2024). Each have reported high performance in diverse tasks including text-based reasoning and represent the sate-of-the-art in generalist LLMs at the time of writing. Specifically, we used the following model versions: `gpt-4o-2024-05-13`, `Meta-Llama-3.1-405B-Instruct-Turbo`, `gemma-2-27b-it`, and `Mixtral-8x22B-Instruct-v0.1`.[3] Temperature is set to 0.0 with all other parameters left at their default values.

**Prompting Strategy.** LLMs are notoriously oversensitive to prompt format (Wadhwa et al., 2023). For the purpose of our analyses, we explored several prompting strategies (listed in Appendix C) and selected the most successful one as measured by correlation with majority-voted human ratings. Most models worked best with *verbatim* prompts corresponding to a word-by-word copy of the guidelines given to humans, to which they responded with a structured list of criteria and their assigned labels for the given target explanation.

**Label Extraction.** Using free-text generation models for a classification task introduces the problem of extracting said ratings, and presents an information extraction challenge in itself. This phenomenon, inherent to generative approaches (Wadhwa et al., 2023), is a source of additional noise that affects all evaluation pipelines necessitating a non-trivial solution in real applications. In our experiments, we used a rule-based extraction method backed up with LLM-based extraction in case of failure. Finally, we manually inspected the remaining failures,[4] and excluded them from our experiments to maintain a fair comparison. The final extraction failure rates were <0.2% for all models but Gemma-2 (2.7%).

## 3.2 Inter-annotator Agreement

In subjective tasks some degree of label variance is expected, resulting in lower inter-annotator agreement. This disagreement is not necessarily noise but a feature of the data, as it can reflect the diversity of human opinions (Aroyo & Welty, 2015). In this context, regardless of absolute agreement, a successful LLM-based rater should be harmonious with the range of human labels rather than deviate from it.

To measure this, we compared the inter-annotator agreement (Krippendorff's $\alpha$) between human raters and when a random rater is replaced by an LLM. There are three possible outcomes: (i) agreement *decreases*, indicating that the LLM deviates from human judgments;

---

[3]We also provide the results of several smaller or older alternatives in the appendix for comparison (Tables 7 and 8).

[4]Mostly due to non-compliance to the task format, e.g., responding verbosely with new labels instead of following the given choice: *"... Related: Somewhat. ..."* (instead of Yes or No)

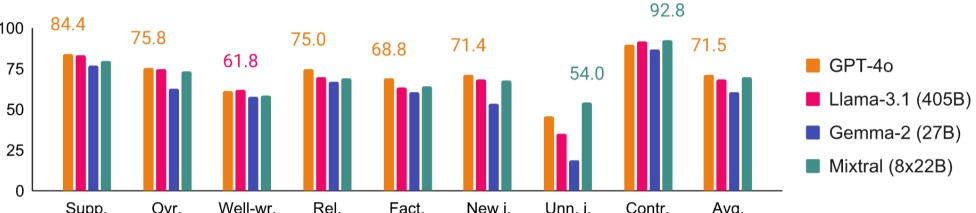

Figure 4: Spearman's ranking correlation between majority-voted human labels and LLM-generated ratings (∗100 for legibility).

(ii) agreement *remains the same*, indicating that it is harmonious with human judgments; or (iii) agreement *increases*, meaning that the LLM is both harmonious and biased towards a majority.[5]

**Results.**   In Figure 3, the shaded area shows the agreement between human raters, while the bars show the agreement when the respective LLM's ratings replace a random human rater (∗100 for legibility). All values are averaged over twenty iterations. Mixtral, GPT-4o, and Llama-3.1 maintained or improved agreement in most cases, with slight decreases in *supports* with Mixtral, *related* with Llama-3.1, and with *unnecessary information* with GPT-4o and Llama-3.1. Gemma-2, on the other hand, decreased agreement in all but three criteria. However, the latter is also significantly smaller than the others, and illustrates the trade-off between model size and performance. Interested readers may refer to Table 7 in the appendix for all results, including several older or smaller models for comparison.

## 3.3   LLMs As A Replacement for Human Evaluation

The larger models maintained or improved inter-annotator in most criteria, suggesting that they do not deviate from an expected range of human ratings. Here, we ask whether they can then *replace* human evaluation. Specifically, we measured the degree to which the models' predictions align with majority-voted human labels.

**Results.**   Figure 4 shows Spearman's rank correlation (∗100 for legibility) between aggregated human labels and LLM predictions. The highest values are annotated for each criterion. A comprehensive table with all results, including several smaller or older models, is available in the appendix (Table 8).

The correlation in *supports* and *contrastive* was particularly strong: 0.84 and 0.93, respectively. The *unnecessary information* criterion, however, stands out with a much lower correlation in all models (0.54 and less). Others ranged from 0.62 to 0.76, indicating a moderately high correlation. Overall, GPT-4o was the best-performing model in five out of seven criteria, with an average correlation of 0.72. Mixtral, the second-best model, followed closely, particularly outperforming GPT-4o in *unnecessary information* and *contrastive*.

From these results, we conclude that the larger models are relatively well-aligned with humans and could be considered effective depending on the use case. However, they are still clearly not a perfect replacement for human raters. Instead, considering the small change in inter-annotator agreement (§3.2), the model could potentially be used as an *additional* rater when human annotation is scarce, which we explore in the next section.

## 3.4   LLMs As An Additional Rater

The results so far hinted towards LLMs acting similarly to an average human rater. Thus, it may seem appealing as an additional data point when human raters are scarce or expensive. Here, we verified this potential by measuring whether using each model as an additional

---

[5]The latter may be desirable in use cases that rely on majority-voted labels as the ground truth, but comes with the trade-off of losing potentially useful label diversity.

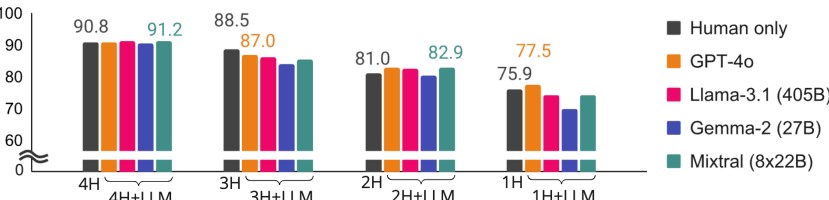

Figure 5: A comparison of Spearman's rank correlation with the original gold labels when using fewer raters (*H) and when an LLM is added as an additional rater (*H+LLM). From left to right, the number of human raters decreases from four to one (randomly selected). All values are multiplied by 100 for legibility.

rater improved the outcome over having fewer human raters, i.e., whether the majority-voted labels became more aligned with the original ones when the LLM was added as a rater.

Specifically, we compared Spearman's rank correlation between the majority-voted labels with all available raters and in two alternative scenarios: one where the model is added as an additional rater and one where it is not. If the correlation *increases* when the model is added, it suggests that its predictions are in line with the original majority-voted labels, and it is useful as an additional rater. Otherwise, it would indicate a harmful or negligible effect, and thus its inclusion should be avoided. We repeated this comparison for scenarios with four, three, two, and one randomly selected human rater per sample.

**Results.**   Figure 5 shows Spearman's rank correlation with the original gold ratings obtained by aggregating the labels of all five raters. Humans only (*H) denotes the correlation between human majority-voted labels only, and others when including the respective LLM as an additional rater (*H+LLM).

Each column cluster represents a different number of human raters from four in the leftmost to one in the rightmost. With four humans, the correlation between their majority-voted labels and the original gold labels was 0.91. Adding a model as a fifth vote raised this by only 0.004 points. With three humans, we observed a slight *decrease* in correlation when adding a model. With two or one human rater, the correlation increased by 0.019 and 0.016 points, respectively.

Overall, the results suggest that LLMs can be useful as additional raters when the number of human raters is less than three. However, even in the best case, the voted labels with an added LLM rater still scored lower than with an equivalent number of human raters (0.83 with 2H+LLM vs. 0.89 with 3H). When there is a high number of human raters, in this case three or more, the model's inclusion as an additional rater does not improve the majority-voted labels' alignment with the original gold labels, and may even harm it.

## 4   Discussion

We analyzed the alignment between human raters and LLMs in the context of evaluating the quality of explanations. The task is highly subjective, and some degree of variance is expected even between humans.

The best-performing models seemed to output labels that are within this variance, suggesting that they behave similarly to individual human raters. However, comparing its outputs to majority-voted labels revealed that their labels are not always in line with human majorities, indicating that they are not suitable as a *complete* replacement for human raters. In turn, when there were fewer human annotators, LLMs helped bring the majority-voted labels closer to the original gold labels. However, this improvement was not seen when there were already three or more human annotators (compared to the original five), suggesting that LLMs are only useful in extremely limited scenarios.

From this, we concluded that LLMs are not a reliable replacement for human raters, unless the use case does not require complete alignment. However, in this work, we operated under the assumption that *majority-voted labels are the ground truth.* In practice, this may not always be the case, and the reliability of LLMs may vary depending on what information is desired in a given scenario. This opens up a new avenue for future work, where LLMs can be applied in a more nuanced manner, considering the context of the task and the intended use of the labels.

Overall, as a balanced approach, we recommend using LLMs in configurations that end with targeted human involvement, such as extensively using LLMs in development or filtering stages and experts for final testing or evaluation. In future work, we plan to explore the behavior of LLMs more closely, particularly in identifying potential patterns in where the model diverged from humans, such as explanations with particular characteristics or in specific contexts. These insights could help us better understand the limitations of LLMs and how they can be used effectively in real-world applications or further improved.

## 5  Related Work

**LLM-based Evaluation**   LLMs as data labelers, and more broadly humans-and-LLMs-in-the-loop settings, are an emerging direction in data collection and labeling. E.g., Wiegreffe et al. (2022) developed a predict-and-explain pipeline that combines GPT-3 with a supervised filter trained on binary acceptability judgments from humans. More recently, Chiang & Lee (2023) proposed using LLMs to evaluate text, closely in line with our work. They, however, found the models successful in their setting. In contrast, we focus on explanation rating with more complex fine-grained criteria, and closely scrutinize potential weaknesses; Previous works brought into question the reliability of LLMs' predictions, especially in prompting setups. E.g., Webson & Pavlick (2022) found that instruction-tuned models often produced good predictions with irrelevant or misleading prompts, bringing into question their real "understanding" of the task. Others reported unreliability of LLMs as labelers in various settings (Albrecht et al., 2022; Shen et al., 2023; Hada et al., 2024, i.a.), a line of work which we join with findings in the context of explanation evaluation.

**Explanation Evaluation**   For explanations in the form of textual justifications, previous works often defined their own criteria for evaluation. Automatic evaluation borrowed from machine learning and measures overlap with "gold standard" text using (a) word-overlap metrics, e.g., BLEU, METEOR and ROUGE; and (b) embedding-based metrics, e.g., BERTScore and BLEURT (Clinciu et al., 2021). In contrast, human-tagged measures are more diverse and explanation-specific. For example, Clinciu et al. (2021) measured *Informativeness* and *Clarity*; Wiegreffe et al. (2022), inspired by social sciences, measured *Acceptability, Generality, Factuality, Grammar, New Info, Supports Label,* and *Amount of Information*; while Aggarwal et al. (2021) defined the criteria of *Refutation, Complete, Comprehensive, Minimal,* and *Coherent*. A recent study instead focused on the *utility* of explanations, i.e., their helpfulness in answering a question from a human point of view (Joshi et al., 2023). In our work, we largely followed these existing criteria, with a modified version of the "*supports*" criterion to capture the *consistency* of a model's predicted label with its justification.

**Explain-and-Predict**   Explanations are often generated in predict-and-explain settings, where models provide justifications for their answers in QA-based benchmarks (e.g., Clinciu et al., 2021), or an elaborate-then-predict setting, where models output a prediction guided by its intermediate outputs such as knowledge statements or reasoning chains (e.g., Maraso-vić et al., 2022; Wang et al., 2023). In this work, we evaluate the capabilities of LLMs to evaluate explanations in an explain-and-predict setting, where models provide justifications for their answers in commonsense reasoning benchmarks. The following sections provide background on each aspect.

# 6 Conclusions

Using a newly-built dataset of free-text explanations and crowdsourced aspect-wise quality ratings, we analyzed the viability of LLMs as explanation evaluators. Stronger LLMs increased or maintained inter-annotator agreement when replacing human ratings, and their ratings were moderately to highly correlated with human ratings, depending on the quality aspect. In a scenario where LLMs were used as additional raters instead of a complete replacement, they improved the outcome when there were only two human raters, but were neutral to detrimental when there are three or more human raters. We conclude that while LLMs can provide ratings moderately consistent with an average human rater, they are not yet reliable enough for complete replacement.

## Limitations

Explanation evaluation is inherently subjective, and the majority-voted gold label is not necessarily the "correct" answer. Subjectivity-informed scoring is a complex task, and we leave its exploration to future work. Furthermore, criteria may change depending on the intended use. Here, we limited our analysis to a researcher's point of view, where explanations are increasingly used as a diagnostic tool, and aligned our criteria to a list of general explanatory competencies. While our insights on LLMs' performance may potentially be applicable to different evaluation tasks, it should not be assumed to be universally true.

LLMs are known to be sensitive to prompt format. We somewhat compensated for this by comparing a prompt-averaged setting, and our focus is not to search for the optimal setting but rather investigate potential fundamental issues with LLM-based evaluation. However, for practical applications, we acknowledge that there is a wealth of tweaks that may improve the performance. Nevertheless, our findings highlight the need for caution when using LLMs for explanation evaluation, and we hope that our work will inspire further research in this direction.

## Ethics Statement

Some of the commercial LLMs we used have built-in filters for potentially harmful content, which were triggered during several of our experiments. This data should not be used in any downstream applications without first controlling for potentially harmful samples.

Crowdworkers play a vital role in dataset creation. We prioritized fair compensation, transparency in data usage, and respect for their rights and privacy. Workers were informed on their work's intended usage and of our identity as requesters. We did not collect any personal information.

## Acknowledgements

This work was supported by JST CREST Grant No. JPMJCR20D2 and JSPS KAKENHI Grant No. 15H01702 and 21K17814. We extend sincere thanks to all the crowdworkers who helped realize this project with their diligent work in labeling the dataset. A big thanks to Tatsuki Kuribayashi and Jonas F. Lotz for their detailed feedback on the paper.

This work was written with the help of LLM-powered writing and coding assistants, however, all ideas and opinions presented in this work are solely those of the authors.

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

## A  Dataset Statistics

Table 2 shows a breakdown of the samples in the test set per source dataset as described in Section 2.2. Table 3 shows the label distributions of each criterion in our test set. Tables 5 and 6 show examples of best-rated and worst-rated explanations from each source dataset.

| Dataset | #samples |
|---|---|
| COPA-SSE (best & worst) | 250 + 250 |
| + fluency fix | 250 |
| Generated (BCOPA) | 500 |
| Crowdsourced (BCOPA) | 500 |
| CoS-E | 500 |
| + fluency fix | 250 |
| Generated (CommonsenseQA) | 500 |
| ECQA | 500 |
| Total | 3,500 |

Table 2: Breakdown of explanation data used in our experiments by source dataset.

| Criterion | -1 | 0 | 1 | 2 | 3 | 4 | 5 |
|---|---|---|---|---|---|---|---|
| Supports | 11% | 32% | 31% | 7% | 9% | 8% | |
| Overall | | | 16% | 13% | 28% | 28% | 12% |
| Well-written | | | 27% | 72% | | | |
| Related | | | 6% | 93% | | | |
| Factual | 10% | 4% | 84% | | | | |
| New Info | | | 30% | 31% | 35% | 2% | |
| Unnecessary Info | | | 82% | 17% | | | |
| Contrastive | | | 58% | 41% | | | |

Table 3: Label distributions per criterion of majority-voted human ratings. -1 denotes "none" for *supports* and "N/A" for *factual*.

## A.1 Dataset-Wise Average Ratings

Table 4 shows the mean ratings and standard deviations of majority-voted ratings per data subset, excluding the categorical aspect *supports*. Data labels are described in Section 2.1. All ratings are the higher the better, except for *unnecessary information* which is the lower the better.

| Data source | Ovr. 1-5 | Well-wr. 0, 1 | Rel. 0, 1 | Fact. -1, 0, 1 | New i. 0-3 | Unn. i. 1, 0* | Cntr. 0, 1 |
|---|---|---|---|---|---|---|---|
| CoS-E | $1.89_{0.95}$ | $0.34_{0.47}$ | $0.77_{0.42}$ | $0.23_{0.95}$ | $0.32_{0.51}$ | $0.41_{0.49}$ | $0.01_{0.09}$ |
| CoS-E + fl. fix | $2.06_{1.01}$ | $0.72_{0.45}$ | $0.72_{0.45}$ | $0.39_{0.90}$ | $0.42_{0.67}$ | $0.44_{0.50}$ | $0.01_{0.09}$ |
| CSQA generated | $3.20_{0.83}$ | $0.97_{0.16}$ | $1.00_{0.04}$ | $0.96_{0.26}$ | $1.00_{0.74}$ | $0.04_{0.19}$ | $0.01_{0.09}$ |
| ECQA | $3.05_{0.89}$ | $0.57_{0.49}$ | $1.00_{0.00}$ | $0.90_{0.31}$ | $1.29_{0.71}$ | $0.11_{0.32}$ | $0.96_{0.19}$ |
| COPA-SSE | $2.38_{1.13}$ | $0.45_{0.50}$ | $0.91_{0.28}$ | $0.58_{0.74}$ | $0.84_{0.80}$ | $0.33_{0.47}$ | $0.02_{0.13}$ |
| COPA-SSE + fl. fix | $2.81_{1.02}$ | $0.82_{0.39}$ | $0.97_{0.18}$ | $0.78_{0.60}$ | $0.85_{0.78}$ | $0.18_{0.38}$ | $0.00_{0.00}$ |
| BCOPA generated | $4.21_{0.79}$ | $1.00_{0.04}$ | $1.00_{0.00}$ | $0.97_{0.20}$ | $1.75_{0.63}$ | $0.00_{0.06}$ | $0.95_{0.21}$ |
| BCOPA crowdsourced | $4.32_{0.76}$ | $0.97_{0.17}$ | $1.00_{0.00}$ | $1.00_{0.00}$ | $1.94_{0.55}$ | $0.01_{0.12}$ | $0.95_{0.21}$ |
| All | $3.07_{1.27}$ | $0.72_{0.45}$ | $0.93_{0.25}$ | $0.75_{0.63}$ | $1.11_{0.87}$ | $0.17_{0.38}$ | $0.42_{0.49}$ |

Table 4: Mean ratings and $_{\text{standard deviations}}$ per data subset. Higher is better for all criteria except for *unnecessary information*, marked with an asterisk (*), where lower is better.

The human raters seemed to find generated explanations to be most well-written on average, however, the higher quality human-written explanations (ECQA, BCOPA crowdsourced) had a higher amount of new information. The generated explanations, in turn, had the least amount of *unnecessary* information. Interestingly, even though they were not contrastive, the generated explanations for CommonsenseQA (CSQA generated) had a higher average overall rating than ECQA explanations which are explicitly contrastive.

A Random Forest Regressor, achieving a mean squared error of 0.37, deemed the most important predictive feature to be *new information* (58%), followed by *factual* (20%), *unnecessary*

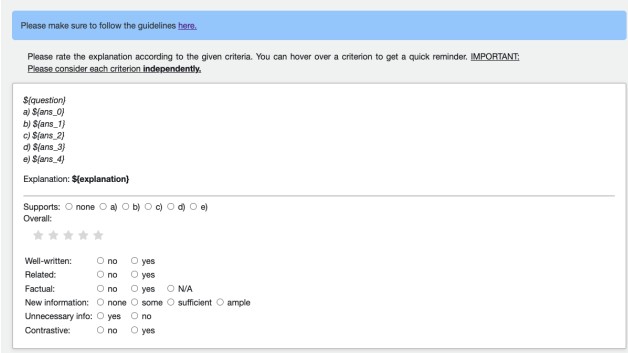

Figure 6: Explanation rating AMT form.

*information* (9%), *well-written* (7%), *contrastive* (4%), *supports* (2%), and *related* (0%). Note that for *supports*, we defined a binary feature of whether the label matches the answer label.

## B  Explanation Rating Annotation

Our crowdsourcing protocol for label collection consisted of three phases: qualification rounds, trial rounds, and main collection rounds. We provided detailed guidelines showing general instructions, detailed information on each criterion and their respective labels, three examples, and a FAQ section based on questions we received from workers. The full document is available upon request to the first author.

**Qualifications**  In the qualification rounds, we curated a question set of 6 explanations and manually tagged them with "acceptable" answers, focusing on overall alignment rather than exact matches. We included a dummy question with strict instructions for filtering. Out of 700 participants, the top 201 workers, with a match percentage of 59% or higher, proceeded to trial rounds. We addressed any concerns or clarifications through email or form feedback. We hand-picked a final group of 28 workers. Qualifications were open to workers with a HIT approval rate of 99% or more and 5,000 or more approved HITs. Note that the location requirement was removed as it was an unnecessary barrier for highly skilled and motivated workers.

**Main Rounds**  Each of the 3,500 explanations in the test set (§2.2) was rated by five workers. The ratings were aggregated to create the final gold labels used in our experiments. Figure 6 shows the crowdsourcing form.

**Payment Information**  For qualifications, each worker was compensated $0.15 per HIT. For the main rounds, the fee was increased to $0.25 per HIT, roughly matching a payment of $20.00 per hour.

## C  Preliminary Experiment: Prompting Strategies

We compared *single* and *compound* calling, where the former prompts the model for a single criterion at a time, and the latter prompts the model for all criteria at once. We also compared *default*, *averaged*, and *verbatim* prompt formats, corresponding to a simple prompt with the explanation and the rating criteria, a voting mechanism over several prompt variants, and an input identical to the human annotation guidelines, respectively. For single *verbatim* calls, only the relevant sections (guidelines and examples) for the target criterion were included. Default and averaged prompts were further compared in zero-shot and three-shot settings, where the latter contained the same examples as shown in the human guidelines. All models were most successful with *verbatim* compound prompting.

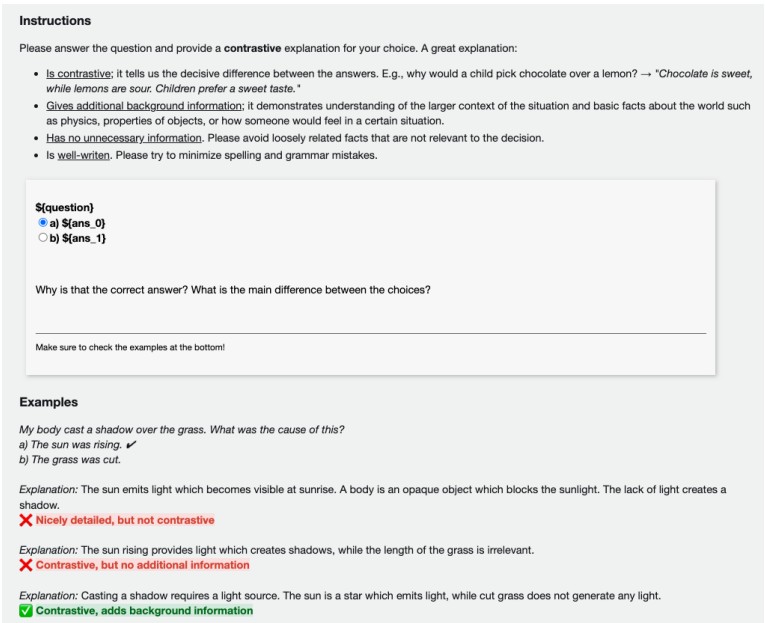

Figure 7: Explanation writing AMT form.

# D  Crowdsourced Explanations for BCOPA

As a supplement to COPA-SSE, We collected 3,000 hand-written, detailed, and contrastive explanations for BCOPA. All crowdsourcing was conducted on the Amazon Mechanical Turk platform.[6] Generally, we found that strict qualifications, hand-picking the final worker pool, and maintaining open communication led to a significant increase in data quality.

**Qualifications**   In the qualification rounds, workers were asked to write three explanations for an explanation, its mirrored sample,[7] and another random explanation. 397 workers were hand-picked and whitelisted for further rounds based on the quality of their responses. We limited participation in this qualification to workers with a HIT approval rate of 99% or more, 1,000 or more approved HITs, and located in Great Britain or the United States.

**Main Rounds**   The workers were explicitly instructed to write *contrastive* explanations which *"focus on what the key difference between the options is, and how it leads to it being the correct choice in one case and not in the other."* This instruction was inspired by insights from Miller (2019). Figure 7 shows the crowdsourcing form. We collected two explanations per BCOPA question, totaling 3,000 explanations. Out of these, a random sample of 500 explanations was rated and included in our test set. Example explanations can be seen in Appendix **??**.

**Payment Information**   For qualifications, each participating worker was compensated $0.10 per HIT. For the main rounds, the fee was increased to $0.89 per HIT, roughly matching a payment of $20.00 per hour.

---

[6]https://requester.mturk.com/

[7]With the same answer choices but a different question that makes the alternative correct.

**CoS-E (best)**

*Which effect of stress could cause death if not treated immediately?*
*a) age grieving person b) **heart attacks** c) depression d) hair loss e) headache*

*Explanation:* heart attacks can stop your heart from properly pumping blood, thus leading to death.

| Over.: | 4 | Well-wr.: | yes |
|---|---|---|---|
| Supp.: | b) | Rel.: | yes |
| | | Fact.: | yes |
| | | New i.: | ++ |
| | | Unn. i.: | no |
| | | Contr.: | no |

**CoS-E (worst)**

*The college kids weren't known for their apartment decor choices, they used a folding chair inside their what?*
*a) bathroom b) closet c) school d) **room** e) church*

*Explanation:* after years of being held captive in an isolated

| Over.: | 1 | Well-wr.: | no |
|---|---|---|---|
| Supp.: | no | Rel.: | no |
| | | Fact.: | n/a |
| | | New i.: | – |
| | | Unn. i.: | yes |
| | | Contr.: | no |

**CoS-E + fluency fixed (best)**

*What kind of driving leads to accidents?*
*a) stressful b) **dangerous** c) fun d) illegal e) deadly*

*Explanation:* Driving dangerously creates accidents.

| Over.: | 4 | Well-wr.: | yes |
|---|---|---|---|
| Supp.: | b) | Rel.: | yes |
| | | Fact.: | yes |
| | | New i.: | + |
| | | Unn. i.: | no |
| | | Contr.: | no |

**CoS-E + fluency fixed (worst)**

*What is a grumpy person likely to have?*
*a) **headache** b) understand themselves c) compliments d) own house e) happiness*

*Explanation:* A peculiar person.

| Over.: | 1 | Well-wr.: | no |
|---|---|---|---|
| Supp.: | none | Rel.: | no |
| | | Fact.: | n/a |
| | | New i.: | – |
| | | Unn. i.: | yes |
| | | Contr.: | no |

**Generated (best)**

*The cloth's are not yet placed where they belong, where should we put them?*
*a) **linen closet** b) kitchen c) on the space heater d) fabric store e) bedroom*

*Explanation:* linen closets are typically used to store items such as cloths.

| Over.: | 5 | Well-wr.: | yes |
|---|---|---|---|
| Supp.: | a) | Rel.: | yes |
| | | Fact.: | yes |
| | | New i.: | ++ |
| | | Unn. i.: | no |
| | | Contr.: | no |

**Generated (worst)**

*Sarah opened his chest and found a second heart. Her patient might not be what?*
*a) istanbul b) animal c) antique shop d) **human being** e) bedroom*

*Explanation:* a human being is the only creature known to have two hearts.

| Over.: | 1 | Well-wr.: | yes |
|---|---|---|---|
| Supp.: | d) | Rel.: | yes |
| | | Fact.: | no |
| | | New i.: | – |
| | | Unn. i.: | no |
| | | Contr.: | no |

**ECQA (best)**

*Where would one find a captive monkey?*
*a) rain forest b) tree tops c) toy store d) **zoological gardens** e) great outdoors*

*Explanation:* A monkey can be kept captive at a zoological garden as many animals are kept there for people to see. All the other options are incorrect as they do not have captive monkeys.

| Over.: | 5 | Well-wr.: | yes |
|---|---|---|---|
| Supp.: | d) | Rel.: | yes |
| | | Fact.: | yes |
| | | New i.: | ++ |
| | | Unn. i.: | no |
| | | Contr.: | yes |

**ECQA (worst)**

*Where are there lots of seats placed in rows surrounding a court?*
*a) show b) **auditorium** c) movies d) soccer stadium e) hockey game*

*Explanation:* In auditorium, there are lots of seats placed in rows. Auditorium seats are surrounding a court. In show, there are no surrounding court placement of seats. Movies is not a place. In soccer stadium, seats are not placed in rows. In hockey game, there are no proper seats.

| Over.: | 2 | Well-wr.: | no |
|---|---|---|---|
| Supp.: | b) | Rel.: | yes |
| | | Fact.: | no |
| | | New i.: | – |
| | | Unn. i.: | no |
| | | Contr.: | yes |

Table 5: Top- and bottom-rated example explanations for CommonsenseQA in 🌰ACORN. New info. labels are denoted as follows: +++ (ample), ++ (sufficient), + (some), – (none).

| | | | | |
|---|---|---|---|---|
| **COPA-SSE (best)**
*The vehicles travelled slowly. Cause?*
a) **It was foggy.**      b) It was cloudy.
*Explanation:* Fog is a thick mist which prevents you from driving quickly as you cannot see far in front of your vehicle. Driving slow through is motivated by the desire not to crash. | Over.:
Supp.: | 5
a) | Well-wr.:
Rel.:
Fact.:
New i.:
Unn. i.:
Contr.: | yes
yes
yes
++
no
no |
| **COPA-SSE (worst)**
*The boy found the television show interesting. Result?*
a) He shut it off.      b) **He recorded it.**
*Explanation:* The result causes the record shut off. | Over.:
Supp.: | 1
none | Well-wr.:
Rel.:
Fact.:
New i.:
Unn. i.:
Contr.: | no
yes
n/a
–
yes
no |
| **COPA-SSE + Fluency fixed (best)**
*The woman tolerated her friend's difficult behavior. Cause?*
a) **The woman knew her friend was going through a hard time.**
b) *The woman felt that her friend took advantage of her kindness.*
*Explanation:* Going through a hard time can cause people to act in a difficult way. A friend can be motivated to support their friend when they're having a tough time. | Over.:
Supp.: | 5
a) | Well-wr.:
Rel.:
Fact.:
New i.:
Unn. i.:
Contr.: | yes
yes
yes
++
no
no |
| **COPA-SSE + Fluency fixed (worst)**
*The shirt did not have a hole anymore. Cause?*
a) **I patched the shirt.**      b) I bleached the shirt.
*Explanation:* I desire to look good. | Over.:
Supp.: | 1
none | Well-wr.:
Rel.:
Fact.:
New i.:
Unn. i.:
Contr.: | no
no
n/a
–
yes
no |
| **Generated (best)**
*The boy was struggling to fall asleep. Cause?*
a) **He turned on his nightlight.**      b) He watched a horror movie.
*Explanation:* Watching a horror movie can create fear and anxiety that can make it difficult for someone to fall asleep. The nightlight is not likely to be the cause since it is meant to help someone sleep. | Over.:
Supp.: | 5
b) | Well-wr.:
Rel.:
Fact.:
New i.:
Unn. i.:
Contr.: | yes
yes
yes
++
no
yes |
| **Generated (worst)**
*The woman awakened. Cause?*
a) She took sleeping pills.      b) **Her husband was snoring.**
*Explanation:* Taking sleeping pills is a direct cause of awakening, whereas a husband snoring is an indirect cause. | Over.:
Supp.: | 1
a) | Well-wr.:
Rel.:
Fact.:
New i.:
Unn. i.:
Contr.: | yes
yes
no
+
no
yes |
| **Crowdsourced (best)**
*The man was bitten by mosquitoes. Cause?*
a) He fell asleep on his couch.      b) **He went camping in the woods.**
*Explanation:* Sleeping on a couch would mean you are indoors where there are rarely mosquitoes. Mosquitoes are prevalent in wooded areas, so the man would be more likely to be camping in the woods, if he was bit by mosquitoes. | Over.:
Supp.: | 5
b) | Well-wr.:
Rel.:
Fact.:
New i.:
Unn. i.:
Contr.: | yes
yes
yes
+++
no
yes |
| **Crowdsourced (worst)**
*I refilled my water bottle. Cause?*
a) **I drank all the water in it.**      b) I kept it in the fridge.
*Explanation:* If you kept your water in the fridge, it would not need to be refilled. | Over.:
Supp.: | 2
a) | Well-wr.:
Rel.:
Fact.:
New i.:
Unn. i.:
Contr.: | yes
yes
yes
+
no
yes |

Table 6: Top- and bottom-rated example explanations for BCOPA in 🐿ACORN. New info. labels are denoted as follows: +++ (ample), ++ (sufficient), + (some), – (none).

ReasonWriter

| Model version | Supp. | Ovr. | W-wr. | Rel. | Fact. | New i. | Un. i. | Cntr. | Avg. |
|---|---|---|---|---|---|---|---|---|---|
| gpt-3.5-turbo-0613 | 0.861 | 0.561 | 0.327 | 0.644 | 0.413 | 0.458 | 0.386 | 0.756 | 0.551 |
| gpt-4-0613 | **0.873** | 0.624 | 0.346 | **0.673** | 0.427 | 0.476 | 0.397 | **0.812** | **0.578** |
| gpt-4o-2024-05-13 | 0.859 | 0.624 | 0.409 | 0.666 | **0.441** | **0.485** | 0.416 | 0.794 | 0.587 |
| gpt-4o-mini-2024-07-18 | 0.852 | 0.619 | 0.411 | 0.648 | 0.423 | 0.474 | 0.393 | 0.791 | 0.576 |
| text-bison-001 | 0.846 | 0.588 | 0.344 | 0.629 | 0.407 | 0.468 | 0.411 | 0.649 | 0.543 |
| gemini-1.0-pro | 0.834 | 0.597 | 0.359 | 0.619 | 0.404 | 0.474 | 0.408 | 0.714 | 0.551 |
| gemma-2-9b-it | 0.836 | 0.553 | 0.360 | 0.634 | 0.403 | 0.407 | 0.368 | 0.748 | 0.539 |
| gemma-2-27b-it | 0.823 | 0.582 | 0.387 | 0.633 | 0.414 | 0.411 | 0.336 | 0.788 | 0.547 |
| Meta-Llama-3.1-8B-*-# | 0.802 | 0.593 | 0.396 | 0.630 | 0.366 | 0.439 | 0.415 | 0.717 | 0.545 |
| Meta-Llama-3.1-70B-*-# | 0.860 | 0.623 | **0.412** | 0.647 | 0.416 | 0.480 | 0.378 | 0.792 | 0.576 |
| Meta-Llama-3.1-405B-*-# | 0.860 | **0.637** | 0.405 | 0.644 | 0.424 | 0.482 | 0.385 | 0.804 | 0.580 |
| Mixtral-8x7B-*-v0.1 | 0.834 | 0.586 | 0.356 | 0.622 | 0.397 | 0.455 | 0.358 | 0.703 | 0.539 |
| Mixtral-8x22B-*-v0.1 | 0.848 | 0.629 | 0.386 | 0.645 | 0.428 | 0.472 | **0.428** | 0.808 | 0.580 |
| Human | 0.866 | 0.613 | 0.365 | 0.655 | 0.399 | 0.442 | 0.433 | 0.784 | 0.570 |

Table 7: Full results of the experiments comparing the difference in inter-annotator agreement between humans and with a random rater replaced by an LLM (§3.2). All values represent Krippendorff's $\alpha$ averaged over 20 iterations. Extraction failures are excluded from analysis. Replace * with "Instruct" and # with "Turbo" in the model names.

| Model version | Supp. | Ovr. | W-wr. | Rel. | Fact. | New i. | Un. i. | Cntr. | Avg. |
|---|---|---|---|---|---|---|---|---|---|
| gpt-3.5-turbo-0613 | 0.846 | 0.627 | 0.409 | 0.696 | 0.621 | 0.611 | 0.383 | 0.786 | 0.622 |
| gpt-4-0613 | **0.900** | 0.740 | 0.515 | **0.778** | 0.675 | 0.691 | 0.528 | **0.946** | **0.722** |
| gpt-4o-2024-05-13 | 0.844 | **0.758** | 0.611 | 0.750 | **0.688** | **0.714** | 0.457 | 0.896 | 0.715 |
| gpt-4o-mini-2024-07-18 | 0.853 | 0.708 | 0.620 | 0.700 | 0.628 | 0.678 | 0.463 | 0.888 | 0.692 |
| text-bison-001 | 0.773 | 0.685 | 0.456 | 0.654 | 0.653 | 0.640 | 0.515 | 0.607 | 0.623 |
| gemini-1.0-pro | 0.787 | 0.681 | 0.497 | 0.613 | 0.549 | 0.671 | 0.451 | 0.648 | 0.612 |
| gemma-2-9b-it | 0.798 | 0.645 | 0.492 | 0.674 | 0.584 | 0.616 | 0.388 | 0.742 | 0.617 |
| gemma-2-27b-it | 0.769 | 0.626 | 0.574 | 0.668 | 0.603 | 0.537 | 0.191 | 0.867 | 0.604 |
| Meta-Llama-3.1-8B-*-# | 0.632 | 0.638 | 0.562 | 0.639 | 0.535 | 0.555 | 0.469 | 0.662 | 0.586 |
| Meta-Llama-3.1-70B-*-# | 0.847 | 0.732 | **0.632** | 0.703 | 0.652 | 0.680 | 0.311 | 0.886 | 0.680 |
| Meta-Llama-3.1-405B-*-# | 0.833 | 0.745 | 0.618 | 0.699 | 0.631 | 0.686 | 0.351 | 0.916 | 0.685 |
| Mixtral-8x7B-*-v0.1 | 0.747 | 0.624 | 0.487 | 0.616 | 0.543 | 0.604 | 0.314 | 0.640 | 0.572 |
| Mixtral-8x22B-*-v0.1 | 0.799 | 0.733 | 0.588 | 0.688 | 0.642 | 0.681 | **0.540** | 0.928 | 0.700 |

Table 8: Full results of the experiments measuring Spearman's rank correlation between majority-voted human labels and LLM-generated ones (§3.3). Extraction failures are excluded from analysis. Replace * with "Instruct" and # with "Turbo" in the model names.

