# OpenReview forum: "ACORN: Aspect-wise Commonsense Reasoning Explanation Evaluation"
_colmweb.org/COLM/2024/Conference — COLM_

### Official Review · Reviewer_e4Ah · 2024-05-09

**Rating:** 8
**Confidence:** 4
**Ethics Flag:** 1

**Summary:**

This paper examines the potential of large language models (LLMs) to automatically evaluate the quality of explanations in commonsense reasoning tasks. The authors introduce ACORN, a novel dataset of 3,500 explanations with human ratings across eight different quality aspects. The authors explore the capabilities of various LLMs to replace or supplement human evaluation, analyzing inter-annotator agreement and correlation with gold-standard labels. They find that LLMs often deviate from human judgments, but GPT-4 shows some potential, particularly as an additional rater when human resources are limited.

**Reasons To Accept:**

S1. The paper thoroughly investigates various LLMs and their impacts on inter-annotator agreement and correlation with human judgments. The experimental design, which examines LLMs as potential replacements for human annotators, was particularly interesting. The results offer insightful ideas on how to use LLMs to complement human evaluations.

S2. ACORN provides a valuable resource for research into automatic explanation evaluation, offering a diverse set of explanations and comprehensive human ratings across varying qualities.

S3. Overall, the paper is well-written and easy to follow. The authors clearly document their data collection and analysis procedures, enhancing the transparency and reproducibility of their findings.

**Reasons To Reject:**

While the paper focuses on a topic that may interest a narrow audience (evaluating explanations), its contributions are clearly and concretely articulated. However, I have one minor concern. The datasets used are of varying quality, yet the analysis encompasses all datasets uniformly. It would have been advantageous if the authors had conducted analyses on subsets of the datasets to determine if LLMs can be more effective with texts of particular quality levels. Additionally, the purpose of including fluency-improved versions in the datasets was not clear. Clarification on this aspect would be helpful.

---

> ### Author Rebuttal · Authors · 2024-05-31
>
> Thank you very much for highlighting the value of our work.
>
> A deeper model performance analysis, such as a quality-wise breakdown, is a great addition. Thank you for the suggestion; we will update our paper to include it. A quick run with GPT-4 revealed that it improved inter-annotator agreement more in lower-rated explanations than in higher-rated ones:
>
> | Overall rating | $\Delta \alpha *100$  |
> |---|---|
> | 1 star | +2.24 |
> | 2 stars | +3.86 |
> | 3 stars | +2.13 |
> | 4 stars | +0.33  |
> | 5 stars | -0.79  |
> | All samples | +0.91 |
>
> (As a reminder, the values represent the difference in inter-annotator agreement when the LLM replaces one random rater out of the five, averaged over 20 iterations and *100 for legibility. The samples are bucketed by the aggregated overall rating given by human annotators. This is related to the first setting, i.e., Section 3.2.)
>
>
> As for why we include fluency-improved versions, we hope our answer to Reviewer RfVm regarding the same question clarifies this point:
>
> > Well-written explanations typically also have high scores in all other aspects. To prevent fluency from becoming a superficial signal, we created fluent but imperfect samples w.r.t. other aspects by improving the fluency of low-quality explanations.

---

### Official Review · Reviewer_RfVm · 2024-05-10

**Rating:** 6
**Confidence:** 4
**Ethics Flag:** 1

**Summary:**

This paper evaluates the utility of LLMs for evaluating AI-generated free-text explanations of model predictions. In particular, the quality of the aggregate ratings is tested in various settings where a random human rater is replaced with a suitably prompted LLM. For the most part, LLMs reduce overall annotator agreement and drag the aggregate label distribution away from the majority vote, but one small exception is the case of GPT-4 breaking ties between two humans.

The contribution of the work consists of some commonsense QA explanation data, a pretty straightforward and sound evaluation paradigm for testing whether LMs add productively on top of human raters, and the results of experiments testing several LLMs.

**Questions To Authors:**

General comment: I find the description of the data sources hard to follow. I don't know how to evaluate claims like "improved version" or "aligning with our criteria for well-formed explanations". These seem like subjective qualitative judgments. When describing the provenance of your data, please describe in concrete, non-subjective terms what was done to create it. Some things are also curious to me & unexplained — why did you have to post-process CoS-E and BCOPA for fluency improvement? Could this have introduced errors? Why were only half of them fluency-improved? Does this not also mean that some of your test examples are correlated, hurting your statistical power later on? A lot of things about the setup here seem messy. Can you please clarify in the paper how you processed the data and why?

Question: did you try few-shot prompting? I found a brief mention of it in Appendix D, but I couldn't figure out how it worked into the experiments and whether you tried it. It seems to me that you should probably be doing the best you can to get good performance out of these models and see if they work. Needing a few human labels for few-shot prompting doesn't weaken the case IMO, as you're already assuming access to some human labels anyway (e.g., when adding an LM to a pool of human raters).

In the ethics statement, you say:
> Some of the datasets used in our research have been found to contain biases that can be further perpetuated through explanations supporting them.

Which datasets? Please clarify this and provide citations if it's the case.

**Reasons To Accept:**

The specific methodology for testing LMs as replacing human raters seems really strong to me, and the discussion of how to interpret these results (i.e., if agreement decreases, remains the same, or increases) was solid. I think this paper serves as a good example for future work evaluating similar kinds of questions.
* Not critical, but I think it'd be interesting & informative to add another bar to Figure 4 corresponding to a single random human, to serve as a baseline. As the human labelers are noisy, they shouldn't have 1.0 correlation with the majority vote, and it'd be nice to see how each model compares with that case.

**Reasons To Reject:**

The paper seriously conflates some important issues around explanation quality. The motivation provided in the introduction is about model trustworthiness, interpretability, compliance with "right to explanation" laws, but all of these use-cases of explanations depend on them being faithful to the model's decision making process, i.e., predictive of model behavior. The evaluations done in this paper have little bearing on faithfulness, as they only assess whether the explanation is coherent, correct, and supports the given answer. While explanation coherence is necessary for faithfulness (as incoherent explanations cannot give much information about how the model behaves), the other aspects being measured are about the quality of the explanation with respect to the _actual reason_ a label is correct. A model's ability to report a reason that supports an answer — even the correct reason — is neither necessary nor sufficient for faithfulness, which is about how the explanation relates to the model's decision making process, and cannot be inferred from a single example alone (see e.g. the discussion of "coherent biased reasoning" in https://arxiv.org/abs/2403.05518). Faithfulness, furthermore, is the aspect that is critical for all of the uses of explainability promoted in the intro. Given all of this, I think the paper is not really about explainability, but about evaluating model performance at a specific task, which is producing correct-with-respect-to-the-real-world explanations for classification decisions. I think an accurate presentation of the work would frame it in this way.

All of that said, this framing issue is not crucial to the contribution of the paper, and the experiments and evaluations all seem pretty reasonable to me. So I don't think this is a dealbreaker or anything, I just strongly urge the authors to rework the motivation to avoid any undue implications about the relevance of this work to model trustworthiness, reliability, interpretability. etc., and ask that the discussion of aspects in Table 1 avoid implying that the "Supports" or "Overall" aspects address faithfulness.

The main reason I can imagine this being rejected is that it's overall a pretty small contribution. I don't begrudge it this necessarily but it doesn't feel like it needed the full space it used. Thankfully, the authors did not unnecessarily pad it out to 9 pages, finishing within 8, but there's still quite a bit of fluff and repetition — I feel the paper could have probably fit just fine in 4 pages, and I think it would've made a great short paper. COLM doesn't have a short paper track, and of course this is the first COLM, so I don't really know how to handle this case. I'm recommending the paper for acceptance, as I think it's sound, but leave it to the meta-reviewer to judge.

---

> ### Author Rebuttal · Authors · 2024-05-31
>
> Thank you for the thoughtful comments.
>
> **Re: Faithfulness.**
>
> Our intention behind mentioning use cases of explanations was to stress their increasing importance and prevalence, arguably increasing the need for automatic evaluation. We completely agree that faithfulness is an important aspect of explanations (We cite Jacovi & Goldberg (2020) in the introduction, the work that popularized the distinction between plausibility and faithfulness.)
>
> We also see how this might give the impression that faithfulness should be a larger factor in our analysis since one could reasonably expect that a "right-to-explanation" grants the right to a faithful, not just plausible-sounding, explanation. We will clarify this point by explicitly mentioning the distinction in the introduction and the limitation that our analysis focuses on plausible explanations.
>
> Our reasoning behind the "supports" criterion, which we will also clarify, is the following: we argue for a minimal condition of faithfulness to be that *the predicted answer and the explanation match.* That is, if an explanation points to a different answer than the system's prediction, it proves that it is unfaithful. Thus, we can use the "supports" label to cross-reference with the prediction and potentially identify inappropriate mismatches.
>
> **Q: Quantitative differences between source datasets?** Due to their respective intended usage, CoS-E and COPA-SSE tolerated surface-level flaws by design. In contrast, ECQA and our collection used guidelines explicitly aligned with human preferences. Dataset-wise Overall ratings, to be added, quantitatively confirm this difference.
>
> **Q: Why fluency-improved versions?** Well-written explanations typically also have high scores in all other aspects. To prevent fluency from becoming a superficial signal, we created fluent but imperfect samples w.r.t. other aspects by improving the fluency of low-quality explanations. Even if new errors were introduced, it would still be advantageous for this purpose.
>
> **Q: Why only half?** We deprioritized labeling the augmented samples from unrealistic distributions to maximize the number of informative samples within a limited annotation budget. Their total number still matches the other subsets.
>
> **Q: Few-shot prompting, maximize performance?** Our experiments are equivalent to three-shot prompting. Other settings we tried, as mentioned in Section 3.1, are listed in Appendix C. Please also see our response to Reviewer RR4D, point 3.

---

### Official Review · Reviewer_RR4D · 2024-05-12

**Rating:** 6
**Confidence:** 4
**Ethics Flag:** 1

**Summary:**

This work studies the problem that whether LLM can serve to evaluate free-text reasoning explanations. First of all this work constructs a dataset with 3.5k examples, and by conduction experiments on this dataset, this paper reveals that the inter-annotator agreement is decreased when LLM-generated ratings are used to replace human ratings. More fine-grained results conclude that while LLMs can provide ratings moderately consistent with an average crowdworker, but they are not reliable enough for complete replacement.

**Questions To Authors:**

1. Why not also compare open-sourced LLMs like Mistral/Llama in this work? The numbers on the open-sourced models are also meaningful.

**Reasons To Accept:**

1. The conclusion in this work can inspire futher works on using LLM for automatic evaluations.

2. A dataset with 3500 examples is constructed and can be used for futher research on this task.

**Reasons To Reject:**

Although this work studies whether LLM can be used for evaluating the quality of free-text explanations, the conclusions in this work are a bit weak. First of all the scale of 3500 examples is not large, and meanwhile, with the conclusion that LLMs are not reliable for explanation evaluation, how LLMs can be used for this task are less discussed in this work. For example, whether evaluating on each criteria and later gathering can help the performance.

---

> ### Author Rebuttal · Authors · 2024-05-31
>
> Thank you for the comments. We will update our paper to clarify the points you raised, and we hope our response will resolve your reservations.
>
> 1. **Dataset size.**
>
> Each sample includes eight aspect-wise quality labels from five annotators, resulting in ***140,000 data points.*** This is much larger than many analytical works. For example, [Xu et al. (2023)](https://aclanthology.org/2023.acl-long.181/) collected 260 expert judgments for long-form QA, and [Wiegreffe et al. (2022)](https://aclanthology.org/2022.naacl-main.47/) collected ratings for 250 samples by three raters over seven quality aspects, i.e., 5,250 data points. Our dataset is large enough to be suitable for fine-tuning applications.
>
> Also, note that the total construction cost was around 8,000 USD. If we count the additional high-quality explanations for BCOPA (3,000 in total), the full value surpasses 10,000 USD, a substantial investment for most.
>
> 2. **How *can* LLMs be used?**
>
> We evaluated the models under the strict condition of replacing humans. However, we still view them as relatively successful and useful where *complete* replacement is not required, such as massive-scale data filtering, flagging data for further analysis, or use in model development stages. Our work quantifies current gaps and shows a way to assess future evaluator setups.
>
> 3. **Other settings, e.g., evaluating each criterion separately?**
>
> We considered several setups, including the suggested one (see "Prompting Strategy" in Section 3.1). We chose the one that yielded the highest correlation with aggregated human ratings to create a controlled setting with reasonably strong performance. For example, in our comparisons, compound/separated average Spearman with *verbatim* prompts (best results) was 0.72/0.71 for GPT-4 and 0.62/0.51 for GPT-3.5. All other settings we considered are listed in Appendix C.
>
> 4. **Open-source models?**
>
> Thank you for the suggestion. We selected the strongest model available to us at the time of writing (GPT-4) as a best-case scenario to eliminate doubt that a better model could alleviate the issues we report. The weaker alternatives were included to contextualize the performance and show the outcome with a more affordable alternative.
>
> We will add open-source models to the camera-ready version. However, our main contribution remains the ***dataset and methodology*** for evaluating ***any*** explanation evaluation system, regardless of the models we feature in the paper.

---

> > ### Comment · Reviewer_RR4D · 2024-06-07
> >
> > Thanks for your response. I am still not convinced by the LLM evaluation. But I'll raise my score a bit.

---

### Decision · Program_Chairs · 2024-07-10

**Decision:**

Accept

**Comment:**

The paper investigates the problem of whether LLM can serve to evaluate free-text reasoning explanations. A dataset called ACORN, containing 3,500 examples with human ratings across eight different quality aspects, is introduced and used for experiments. They find that LLMs often deviate from human judgments, but GPT-4 shows some potential, particularly as an additional rater when human resources are limited.

Pros:
- The introduction of the dataset with 3,500 examples is valuable, and can be used in future research.
- The paper is well-written, clear, and easy to follow, with transparent and reproducible data collection and analysis procedures.
- The methodology is thorough and well-executed.

Cons:
- Some reviewers expressed concerns about the paper's narrow scope.